# Enhanced Catalytic Oxidation of Toluene over Hierarchical Pt/Y Zeolite

Min-Ryeong Kim [1,2] and Suhan Kim [1,*]

1   Green Materials & Processes R&D Group, Korea Institute of Industrial Technology, 55 Jongga-ro, Jung-gu, Ulsan 44413, Korea; bkst7v@kitech.re.kr
2   Department of Chemical and Biomolecular Engineering, Yonsei University, 50 Yonsei-ro, Seodaemun-gu, Seoul 03722, Korea
*   Correspondence: suhankim@kitech.re.kr

**Abstract:** The development of efficient Pt-supported zeolite catalysts with tunable micro/mesopore structures for the removal of volatile organic compounds (VOCs) presents a major challenge. Herein, hierarchical Pt/Y zeolites with tunable mesopores are fabricated by varying the etching time before the surfactant-templated crystal rearrangement method and used as catalyst supports for VOC oxidation. The hierarchical Pt/Y zeolites provided an excellent environment for Pt nanoparticle loading with abundant accessible acidic sites. The catalytic performance of the obtained hierarchical Pt/Y zeolites is analyzed using toluene oxidation, with the modified zeolites exhibiting improved catalytic activities. The hierarchical Pt/Y zeolites exhibited higher catalytic toluene oxidation activities than non-hierarchical Pt/Y zeolites. Pt/Y-6h demonstrated the highest catalytic toluene oxidation activity of the prepared catalysts, with a $T_{90}$ of 149 °C, reaction rate of $1.15 \times 10^{-7}$ mol $g_{cat}^{-1}$ $s^{-1}$, turnover frequency of $1.20 \times 10^{-2}$ $s^{-1}$, and an apparent activation energy of 66.5 kJ $mol^{-1}$ at 60,000 mL $g^{-1}$ $h^{-1}$ at a toluene concentration of 1000 ppm. This study will facilitate the fine-tuning of hierarchically porous materials to improve material properties and achieve higher catalytic performance toward VOC oxidation.

**Keywords:** hierarchical Pt/Y zeolite; toluene oxidation; tunable mesopore; surfactant-templated; heterogeneous catalysis



## 1. Introduction

Volatile organic compounds (VOCs), the main cause of air pollution, are directly harmful to the human body and must therefore be regulated and controlled [1]. Various methods to eliminate VOCs have been proposed and developed, including adsorption [2], catalytic oxidation [3], bio-treatment [4], photocatalytic removal [5], membrane technology [6], and non-thermal plasma oxidation [7]. Among these technologies, catalytic oxidation is widely used owing to its environmental sustainability and relatively low energy consumption [3,8–10]. Several studies have been conducted that aim to enhance the performance of catalysts toward the oxidation of VOCs by supporting active noble metals, including Pd, Pt, and Au [11–13]. In particular, Pt, a representative noble metal, exhibits high catalytic activity and resistance toward deactivation for the oxidation of VOCs at relatively low temperatures. Zhang et al. [14] reported a $Pt/TiO_2$ catalyst for VOC oxidation and confirmed the effective catalytic activity of the Pt sites. The size of the noble metal particles typically has a significant effect on the catalytic performance of such materials. Several studies have demonstrated that the optimal particle size of Pt for toluene oxidation is approximately 1.9 nm, which ensures high Pt dispersion and a high proportion of $Pt^0$ species [15,16].

Choosing a suitable support is equally as important as using noble metals. Supports with a high specific surface area and large number of mesopores enhance the dispersion of Pt and further improve the accessibility of the active site, thereby reducing the residence

time of the reaction product. Zeolites are extensively used as support materials owing to their high specific surface areas, abundant acidic sites, and high thermal stability [12,17–19]. However, the micropores of the zeolites severely limit the accessibility of the active sites centered in the pores, limiting the catalytic performance of the zeolite [20,21]. This problem can be overcome using a hierarchical zeolite, which is obtained by introducing mesopores into a conventional zeolite as a support. Chen et al. [22,23] showed that Pt-supported hierarchical mesoporous Beta and ZSM-5 zeolites, prepared by dealumination using acid etching, exhibit higher stability and activity for toluene oxidation than conventional Beta and ZSM-5 zeolites. Introducing mesopores into the existing zeolites can improve the accessibility of toluene and accelerate Pt dispersion. Although simple acid etching can increase the mesopore volume by adjusting the etching conditions, the mesopore size cannot be controlled as it increases with an increase in the pore volume. Thus, optimal Pt nanoparticles cannot be obtained. Effective methods to control the micro/mesopores of zeolites and thus obtain desirable mesopore sizes must therefore be developed to support Pt to achieve high catalytic activity for VOC oxidation.

In this study, we developed an effective method to fabricate hierarchical Pt/Y zeolites with tunable mesopores by varying the etching time before implementing the surfactant-templated crystal rearrangement method. Applying a mild acid etching agent induced a slow etching rate was induced, which inhibited the formation of large mesopores. Moreover, the etched mesopores of suitable sizes for Pt loading were obtained using surfactant-templated crystal rearrangement. The large volume of size-controlled mesopores in the prepared hierarchical zeolites induced the formation of small Pt nanoparticles and improved the dispersion and oxidation state of Pt, thereby enhancing the catalytic performance of the hierarchical Pt/Y zeolites for toluene oxidation.

## 2. Results and Discussion

### 2.1. Characterization of the Prepared Catalyst

The obtained powder X-ray diffraction (PXRD) patterns of the Pt/Y zeolites are illustrated in Figure 1. The peak intensities of the hierarchical Pt/Y zeolites did not differ significantly from those of the non-modified Pt/Y zeolite; all catalysts exhibited distinct characteristic peaks for FAU zeolites [24]. The sharp diffraction peaks at 6.2°, 10.2°, 11.9°, 15.7°, 18.7°, 20.4°, 23.7°, 27.1°, 29.7°, 30.8°, 31.4°, 32.5°, 34.1°, and 37.9° correspond to the (1 1 1), (2 2 0), (3 1 1), (3 3 1), (5 1 1), (4 4 0), (5 3 3), (6 4 2), (7 3 3), (6 6 0), (5 5 5), (8 4 0), (6 6 4), and (6 6 6) lattice planes, respectively, clearly identifying these catalysts as highly pure and crystalline cubic zeolites. This finding suggests that the intrinsic crystalline structures of the Y zeolites are retained without significant damage to the crystallinity after the etching and surfactant-templated crystal rearrangement processes. However, the crystallinity is strongly affected by the degree of acid or base treatment [25]. Y zeolites could be acid-etched for a maximum of 6 h without damage to their intrinsic crystal structure, whereas further etching treatment and surfactant-templating affected the Y zeolites framework, thereby reducing the crystallinity and porosity; the details of the etching were reported in a previous study [26]. In addition, the PXRD peaks corresponding to Pt (39.8° and 46.2°) were unclear, implying an even dispersion of the Pt particles in the hierarchical Y zeolites [17,27].

Figure 2a shows the $N_2$ adsorption–desorption isotherms of the Pt/Y zeolites, translated along the *y*-axis to compare their isotherm types. The non-hierarchical Pt/Y zeolite demonstrated a typical type-I isotherm typical of a microporous structure, while the hierarchical Pt/Y zeolites showed a type-IV isotherm and H4 type hysteresis loop associated with mesopores. The increase in the area of the hysteresis loop with etching time characterized the efficiency of this method for the controlled introduction of mesopores in Y zeolites [28]. A relatively large pore size distribution is observed in the hierarchical Pt/Y zeolites (Figure 2b), indicating the presence of a mixture of micropores and mesopores. The detailed textural data for the Pt/Y zeolites is tabulated in Table 1. We prepared hierarchical Pt/Y zeolites using the surfactant-templated crystal rearrangement method, in

which large mesopores are formed through mild acid etching by destroying some micropores in Y zeolites before being rearranged into controlled mesopores by the surfactant templating method. By varying the etching time, the ratio of micropores to mesopores can be changed. Thus, the surface areas of hierarchical Pt/Y zeolites decreased as the pore size changed from microporous to mesoporous. The hierarchical Pt/Y zeolites had larger mesopore volumes (0.19 cm$^3$ g$^{-1}$, 0.19 cm$^3$ g$^{-1}$, and 0.2 cm$^3$ g$^{-1}$) than the non-hierarchical Pt/Y zeolite (0.12 cm$^3$ g$^{-1}$). Among the prepared catalysts, Pt/Y-6h showed the highest mesopore volume.

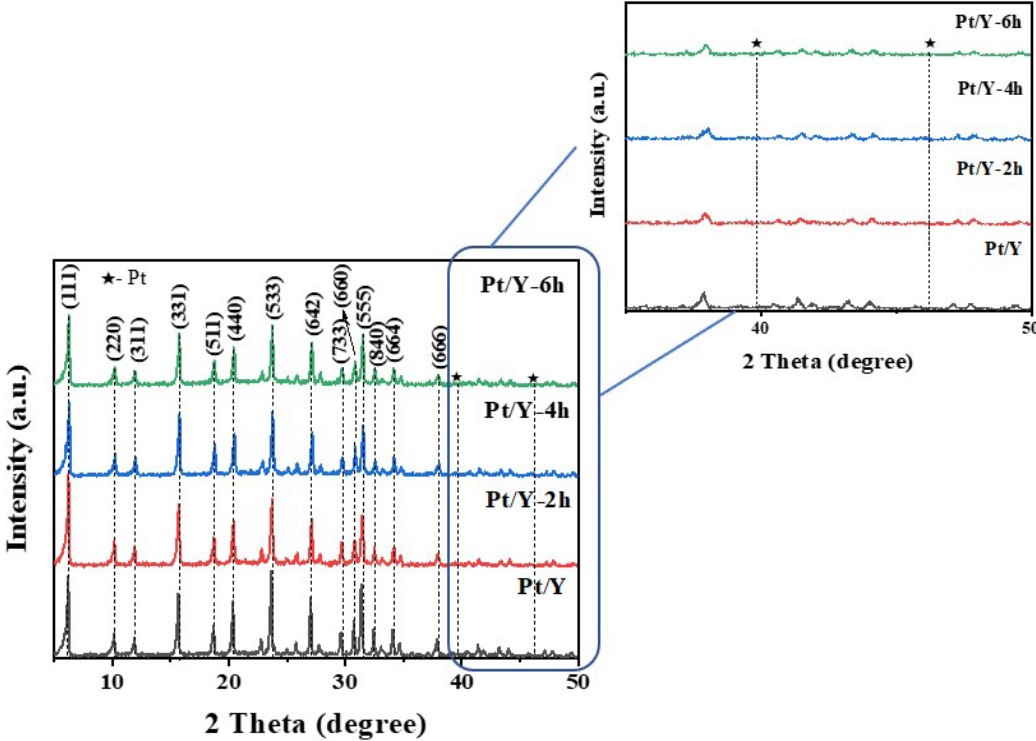

**Figure 1.** Powder X-ray diffraction patterns of the Pt/Y zeolites. The asterisks indicate Pt peaks.

**Table 1.** Textural properties of the Pt/Y zeolites.

| Catalysts | $S_{BET}$ [a] $(m^2\,g^{-1})$ | $V_{pore}$ [b] $(cm^3\,g^{-1})$ | $V_{micro}$ [c] $(cm^3\,g^{-1})$ | $V_{meso}$ [d] $(cm^3\,g^{-1})$ | $V_{meso\ rate}$ (%) |
|---|---|---|---|---|---|
| Pt/Y | 685 | 0.39 | 0.27 | 0.12 | 30.8 |
| Pt/Y-2h | 566 | 0.38 | 0.19 | 0.19 | 50.0 |
| Pt/Y-4h | 529 | 0.37 | 0.18 | 0.19 | 51.4 |
| Pt/Y-6h | 536 | 0.37 | 0.17 | 0.20 | 54.1 |

[a] Determined by the Brunauer–Emmett–Teller (BET) method. [b] Total pore volume of the Pt/Y zeolites adsorbed at $P/P_0 = 0.995$. [c] Analyzed by the t-plot method. [d] Calculated by $V_{pore} - V_{micro}$.

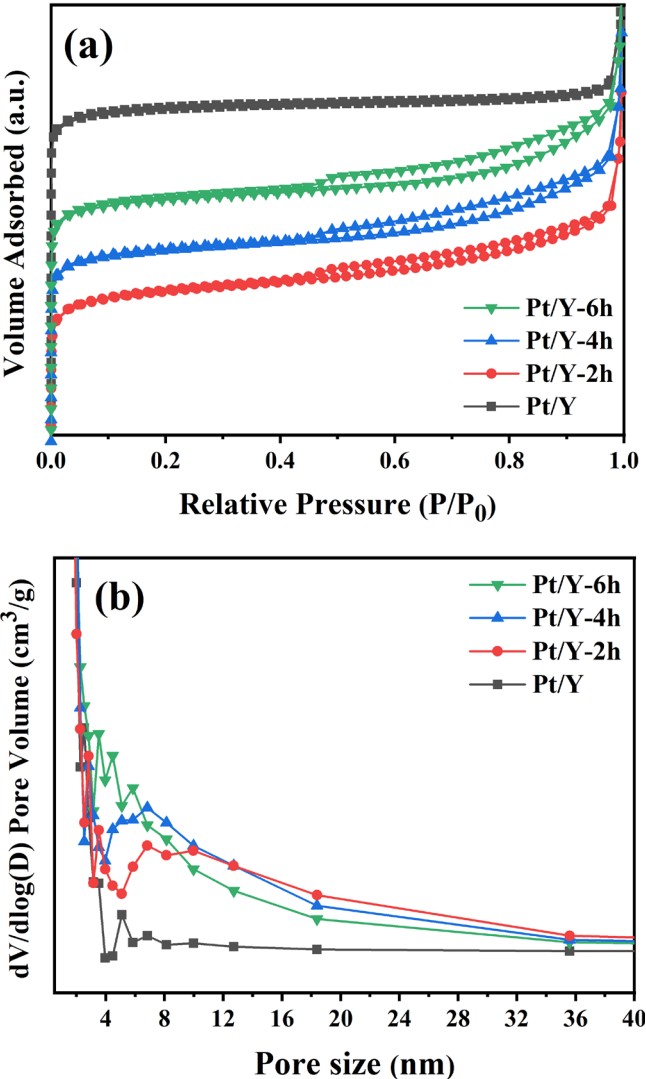

**Figure 2.** (**a**) $N_2$ adsorption–desorption and (**b**) pore size distribution according to the Barrett−Joyner−Halenda (BJH) curves of the Pt/Y zeolites.

Transmission electron microscopy (TEM) images were collected to analyze the morphological changes in the Pt/Y zeolites and sizes of the supported Pt nanoparticles. The Pt nanoparticle sizes, dispersions, loadings, and $Pt^0$ ratios in the Pt/Y zeolites are listed in Table 2. A regular arrangement of several micropores corresponding to the (111) plane, with a d-spacing of 1.43 nm, were observed in the non-hierarchical Pt/Y zeolite (Figure 3a). The TEM images of the hierarchical Pt/Y zeolites made using different etching times are illustrated in Figure 3b–d. Figure 3b demonstrates the coexistence of micropores and mesopores in Pt/Y-2h. In contrast, Pt/Y-4h and Pt/Y-6h exhibit only mesopores, as shown in Figure 3c,d, respectively. Nevertheless, the pore size distribution curves revealed the existence of micropores in all hierarchical Pt/Y zeolites that were difficult to detect on the material surface owing to their presence within the Y zeolites. The average sizes of the Pt nanoparticles in Pt/Y, Pt/Y-2h, Pt/Y-4h, and Pt/Y-6h were approximately 11.5, 9.3, 4.3, and 2 nm, respectively, according to the TEM results. Pt is evenly dispersed in Pt/Y-6h (Figure 3d), whereas aggregation of Pt is observed in Pt/Y (Figure 3a). The dispersion of the Pt measured by CO chemisorption analysis is consistent with the TEM results, with Pt/Y-6h demonstrating the most significant CO uptake among the catalysts. The Pt dispersion in Pt/Y-6h (45.5%) is approximately six times greater than that in Pt/Y (7.7%). Moreover, the Pt dispersion decreases significantly with increasing particle size [29–31].

The particle size and dispersion of Pt influences the catalytic activity. Hence, they should be appropriately adjusted. Thus, introducing mesopores into Y zeolites allows the structure of the Pt nanoparticles to be controlled, as reported in previous studies [28,32–35].

**Table 2.** Transmission electron microscopy (TEM), X-ray photoelectron spectroscopy (XPS), inductively coupled plasma-atomic emission spectrometry (ICP-AES), and CO chemisorption analysis results for the Pt/Y zeolites.

| Catalysts | Pt Particle Size [a] (nm) | Pt Particle Size [b] (nm) | $Pt^0/(Pt^0 + Pt^{2+})$ [c] (%) | Pt Loading [d] (%) | CO Uptake [b] (μmol g$^{-1}$) | Pt Dispersion [b] (%) |
|---|---|---|---|---|---|---|
| Pt/Y | 11.5 | 12.3 | 40 | 0.39 | 1.53 | 7.7 |
| Pt/Y-2h | 9.3 | 8.5 | 58 | 0.50 | 2.52 | 13.3 |
| Pt/Y-4h | 4.3 | 5.0 | 64 | 0.37 | 4.81 | 18.8 |
| Pt/Y-6h | 2.0 | 2.0 | 74 | 0.41 | 9.57 | 45.5 |

[a] Calculated using the TEM images. [b] Measured by CO chemisorption analysis. [c] Calculated using the XPS profiles. [d] Measured by ICP-AES.

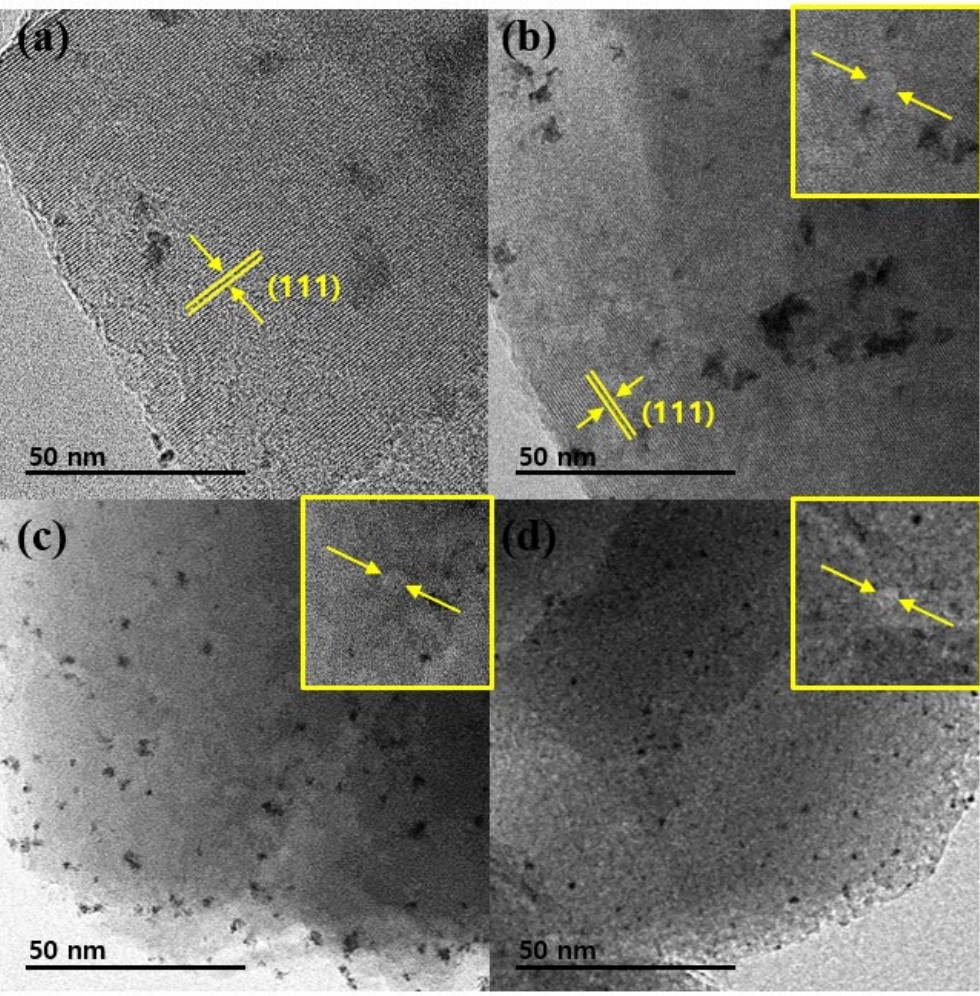

**Figure 3.** Transmission electron microscopy (TEM) images of the Pt/Y zeolites (**a**) Pt/Y, (**b**) Pt/Y-2h, (**c**) Pt/Y-4h, and (**d**) Pt/Y-6h. The arrows indicate mesopores.

The deconvoluted Pt 4*f* X-ray photoelectron (XPS) spectra of the Pt/Y zeolites are shown in Figure 4. Since the Al 2*p* peaks overlapped with the Pt 4*f* peaks in the 68–80 eV range, the Al 2*p* peak was deconvoluted and assigned to 73.9 eV The peaks at 70.1 eV and

73.4 eV were assigned to the $Pt^0$ species, while those at 71.0 eV and 74.3 eV were assigned to the $Pt^{2+}$ species [36,37]. The $Pt^0/(Pt^0 + Pt^{2+})$ ratios are listed in Table 2. The fraction of Pt/Y-6h (74%) was higher than that of Pt/Y (40%) owing to the size and dispersion for the Pt nanoparticles [23,30]. Smaller and more highly dispersed particles interact more strongly with their zeolite supports, resulting in a higher concentration of the surface $Pt^0$ species. Conversely, larger particles with low dispersion interact weakly with the zeolite support, resulting in a lower concentration of the surface $Pt^0$ species. This $Pt^0$ species plays an important role in the catalytic performance since it is the active species that completely oxidizes toluene to $CO_2$ and $H_2O$. In particular, Pt/Y-6h, which is rich in $Pt^0$ species, exhibited outstanding catalytic activity.

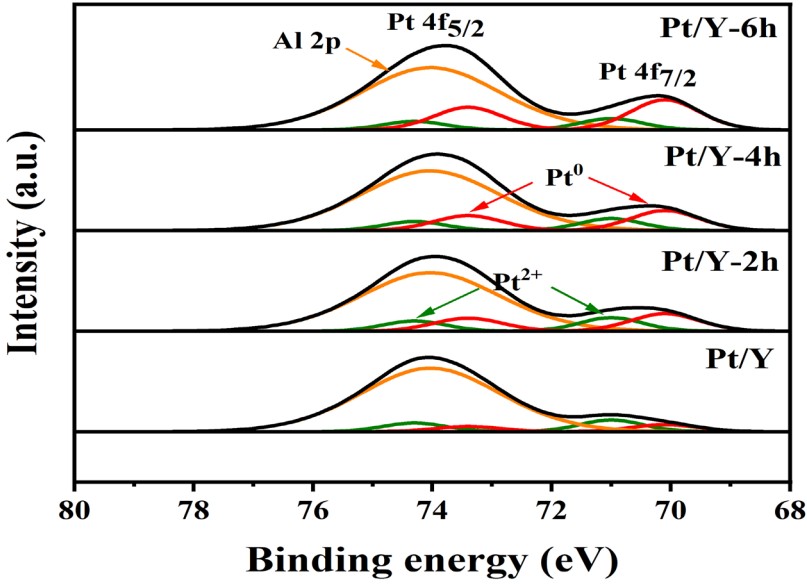

**Figure 4.** Deconvoluted Pt 4*f* X-ray photoelectron spectra of the Pt/Y zeolites.

Temperature-programmed desorption of ammonia ($NH_3$-TPD) was performed to measure the acidic properties of the Pt/Y zeolites (Figure 5). Surface acidic sites are known to be strongly related to catalytic activity. The adsorption of toluene increases with the number of acidic sites, thereby increasing the surface coverage of toluene on the catalyst and facilitating the catalytic oxidation reaction [38]. The low temperature (100–250 °C) ammonia desorption peak in the $NH_3$-TPD curve corresponds to the weak acid sites, the high temperature (400–600 °C) peak corresponds to the strong acid sites, and the intermediate temperature (250–400 °C) peak corresponds to the medium acid sites [39]. Table 3 presents the number of acidic sites represented by each peak. The main ammonia desorption peak of the non-hierarchical zeolite appears at approximately 164 °C. In contrast, the hierarchical Pt/Y zeolites showed an increased amount desorbed ammonia, with the main peaks of Pt/Y-2h, Pt/Y-4h, and Pt/Y-6h appearing at 181, 179, and 242 °C, respectively. These observations confirm that the number of acidic sites in hierarchical Pt/Y zeolites increased, and that the acidity of such zeolites is also strengthened compared to the non-hierarchical Pt/Y zeolite. As the etched surface area increases during etching, the number of acidic sites also increases owing to an increase in the exposure of framework aluminum on the surface. Additionally, if NaY reacts with $NH_4OH$ during the surfactant-templated crystal rearrangement after etching, Na+ is exchanged with $NH_4^+$ during the templating process to form $NH_4Y$, which is then converted to HY during CTAB calcination at 550 °C by elimination of $NH_3$, which is strongly acidic [40].

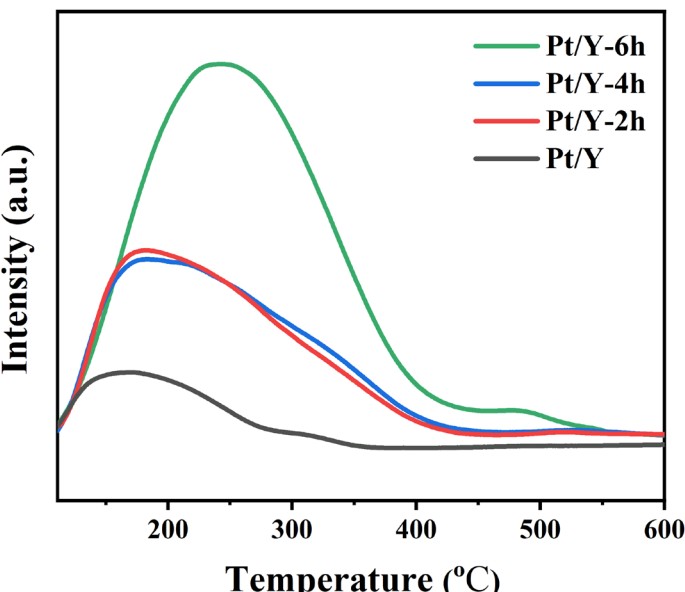

**Figure 5.** Temperature-programmed desorption of ammonia curves of the Pt/Y zeolites.

**Table 3.** Acidic sites and peak temperatures of the Pt/Y zeolites.

| Catalysts | Peak Temperature (°C) | | | Acidic Sites (μmol/g) | | | |
|---|---|---|---|---|---|---|---|
| | Peak I | Peak II | Peak III | Weak | Medium | Strong | Total |
| Pt/Y | 164 | 234 | - | 6 | 4 | - | 10 |
| Pt/Y-2h | 181 | 264 | - | 11 | 16 | - | 27 |
| Pt/Y-4h | 179 | 266 | - | 10 | 20 | - | 30 |
| Pt/Y-6h | 242 | 270 | 450 | 43 | 21 | 7 | 71 |

*2.2. Catalyst Evaluation*

The catalytic activities of the Pt/Y zeolites toward toluene oxidation at different temperatures are shown in Figure 6a, and the $T_{10}$, $T_{50}$, and $T_{90}$ values are presented in Table 4. The toluene oxidation curves of the hierarchical Pt/Y zeolites shifted to lower temperatures than that of the non-hierarchical Pt/Y zeolite and exhibited excellent catalytic activities. The catalytic activity increased with etching time. Pt/Y-6h exhibited the highest catalytic activity for toluene oxidation with a $T_{90}$ of 149 °C. In addition, the $CO_2$ selectivity was very close to 100% regardless of the catalyst and toluene conversion. The generated Arrhenius plots of toluene oxidation by the Pt/Y zeolites are illustrated in Figure 6b. The reaction rate was measured under conditions in which the toluene conversion was less than 20% to remove the influence of mass and heat transfer. The reaction rates, turnover frequency (TOF) values, and apparent activation energies ($E_a$) of toluene oxidation by the Pt/Y zeolites obtained from the Arrhenius plots are summarized in Table 4. The reaction rate and TOF of Pt/Y-6h were $1.15 \times 10^{-7}$ mol $g_{cat}^{-1}$ $s^{-1}$ and $1.20 \times 10^{-2}$ $s^{-1}$, respectively. Both values were the highest among the Pt/Y zeolites. In addition, the $E_a$ of Pt/Y-6h was the lowest at 66.5 kJ mol$^{-1}$. Thermogravimetric analysis (TGA) was performed to calculate the amount of coke deposition on the Pt/Y and Pt/Y-6h after the reaction. Comparatively, coke formation of Pt/Y-6h (0.34%) was much less than that of Pt/Y (1.47%), confirming that Pt/Y-6h has better anti-coking ability.

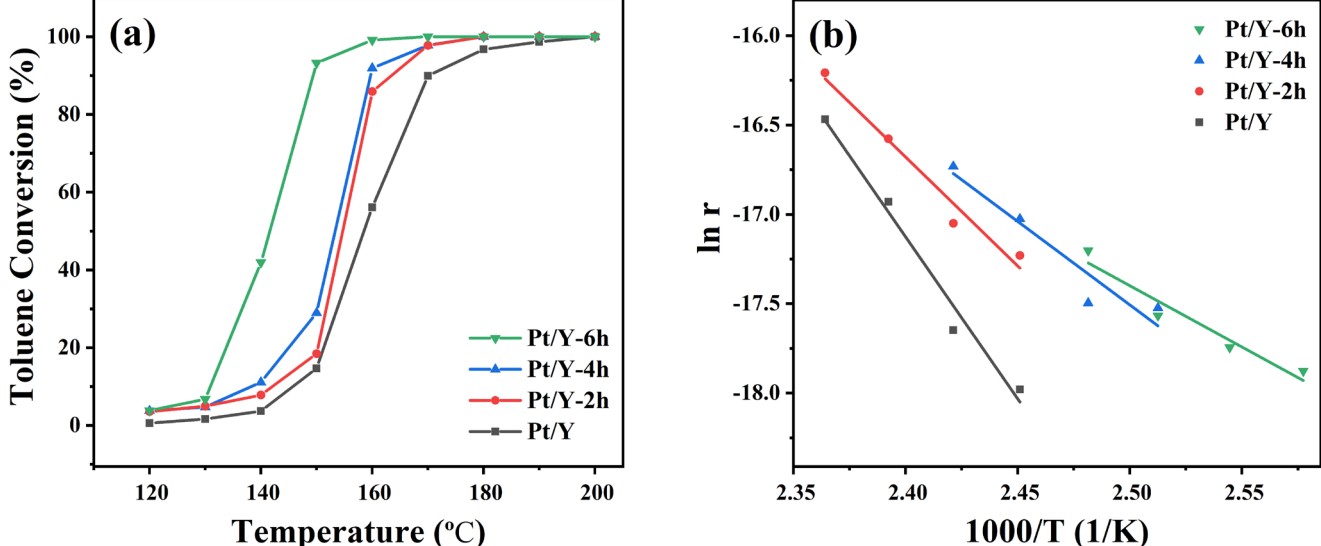

**Figure 6.** (**a**) Dependence of toluene conversion on temperature in the toluene oxidation reaction using the Pt/Y zeolites (the toluene concentration and gas hourly space velocity were 60,000 mL $g^{-1}$ $h^{-1}$, respectively). (**b**) Arrhenius plots of the catalytic oxidation of toluene using the Pt/Y zeolites. The reaction rates were obtained from toluene conversions below 20% in order to eliminate the effects of heat and mass transfer.

**Table 4.** Catalytic performance of the Pt/Y zeolites used in the toluene oxidation reaction.

| Catalysts | Activities | | | $E_a$ [a] (kJ $mol^{-1}$) | Reaction Rates [b] (mol $g_{cat}^{-1}$ $s^{-1}$) | TOF [b] ($s^{-1}$) |
|---|---|---|---|---|---|---|
| | $T_{10}$ (°C) | $T_{50}$ (°C) | $T_{90}$ (°C) | | | |
| Pt/Y | 145 | 159 | 170 | 149.6 | $1.55 \times 10^{-8}$ | $1.10 \times 10^{-2}$ |
| Pt/Y-2h | 142 | 154 | 163 | 99.7 | $3.29 \times 10^{-8}$ | $0.96 \times 10^{-2}$ |
| Pt/Y-4h | 138 | 153 | 159 | 74.8 | $4.04 \times 10^{-8}$ | $1.13 \times 10^{-2}$ |
| Pt/Y-6h | 130 | 141 | 149 | 66.5 | $1.15 \times 10^{-7}$ | $1.20 \times 10^{-2}$ |

[a] The apparent activation energy ($E_a$) was determined by varying the temperature between 115 °C and 150 °C.
[b] The reaction rate and TOF were calculated at 135 °C.

## 3. Materials and Methods

### 3.1. Catalyst Preparation

Hierarchical Pt/Y zeolites were sequentially synthesized from an acidic solution of ammonium hydrogen difluoride ($NH_4HF_2$, Aldrich, St. Louis, MO, USA 95%) and alkaline solution of ammonia ($NH_4OH$, Aldrich, 32%) containing hexadecyltrimethylammonium (CTAB, Aldrich, 98%) as a surfactant. NaY zeolites (5 g, CBV 100, Si/Al = 2.6, Alfa Aesar, Haverhill, MA, USA) were added to 0.2 M $NH_4HF_2$ (100 mL) at 25 °C and stirred at 400 rpm 1 h. The resulting mixture was placed in a Teflon-lined hydrothermal reactor and heated in an oven at 95 °C for 2, 4, and 6 h (samples Y-2h, Y-4h, and Y-6h, respectively). Subsequently, the product was filtered, washed five times with deionized water, and dried in an oven at 90 °C under vacuum for 20 h. CTAB (2 mmol) was dissolved in 0.5 M $NH_4OH$ (50 mL), and the etched sample (1.5 g) was sequentially added at 25 °C while stirring at 400 rpm for 3 h. The mixture was placed in a Teflon-lined hydrothermal reactor and heated in an oven at 150 °C for 12 h. The product was filtered, washed five times with deionized water, and dried in an oven at 90 °C under vacuum for 12 h. To form pores, CTAB was removed by calcining the dried samples in a muffle furnace at 550 °C for 4 h. Finally,

each sample (1 g) was suspended in a 0.4 wt.% aqueous solution of chloroplatinic acid hydrate ($H_2PtCl_6 \cdot xH_2O$, Aldrich, 99.9%) to incorporate Pt in the hierarchical Y zeolites. Subsequently, sodium borohydride ($NaBH_4$, Aldrich, 98%) was carefully added to the mixture at 25 °C with vigorous stirring at 500 rpm for 3 h. The product was filtered, washed five times with deionized water, and dried in an oven at 90 °C under vacuum for 20 h to yield Y-zeolite-supported Pt catalysts. Based on the support labels, the prepared catalysts were labeled as Pt/Y, Pt/Y-2h, Pt/Y-4h, and Pt/Y-6h.

### 3.2. Catalyst Characterization

PXRD was conducted to confirm the crystalline structure of the Pt/Y zeolites. The PXRD patterns were collected using a D/MAX-2500 V Rigaku X-ray diffractometer (Rigaku, Tokyo, Japan) at $2\theta = 5°$–$50°$ at a scan rate of $4°$ $min^{-1}$. The $N_2$ adsorption–desorption isotherms of the Pt/Y zeolites were obtained using an ASAP 2020 Micrometrics instrument. Prior to analysis, the Pt/Y zeolites were pretreated at 300 °C under vacuum for 20 h. Brunauer–Emmett–Teller and *t*-plot analyses were used to determine the specific surface area and micropore volumes of the Pt/Y zeolites, respectively. The pore size distribution was analyzed using the Barrett–Joyner–Halenda method. The morphologies of the Pt/Y zeolites were determined by spherical aberration-corrected scanning transmission electron microscopy (CS-STEM, JEM-ARM300F, JEOL, Tokyo, Japan). The Pt contents of the Pt/Y zeolites were determined by inductively coupled plasma-atomic emission spectrometry (Optima 4300DU, PerkinElmer, Waltham, MA, USA). XPS (Thermo Scientific K-Alpha, Thermo Fisher Scientific, Waltham, MA, USA) was performed to measure the chemical compositions and surface element distributions of the Pt/Y zeolites. An AutoChem II 2920 Micromeritics instrument was employed to perform the $NH_3$-TPD and CO pulse chemisorption experiments. $NH_3$-TPD was conducted to analyze the intensity and concentration of acidic sites on the Pt/Y zeolites. Prior to the $NH_3$-TPD measurements, the Pt/Y zeolites were pretreated using helium gas at 400 °C for 1 h and cooled to 100 °C. Subsequently, the Pt/Y zeolites were saturated with 10% $NH_3$/He gas at 100 °C for 40 min and swept using helium gas for 1 h to extract the physically adsorbed $NH_3$. Finally, the temperature of the Pt/Y zeolites was raised from 100 °C to 650 °C at a rate of 10 °C $min^{-1}$ under helium gas. The dispersion of the Pt catalytic sites in the Pt/Y zeolites was assessed by CO chemisorption after pretreatment using a 10% $H_2$/Ar gas at 450 °C for 1 h. After the pretreatment, the Pt/Y zeolites were purged with helium gas for 40 min, and CO was repeatedly added to reach adsorption saturation. The CO uptake was analyzed using a thermal conductivity detector. TGA (Q500, TA Instruments, New Castle, DE, USA) was performed to confirm the amount of coke deposition for the Pt/Y and PT/Y-6h after catalytic performance test. Data were collected with a heating rate of 5 °C $min^{-1}$ to the final temperature 700 °C under air flow.

### 3.3. Catalyst Performance Evaluation

The catalytic performance was evaluated in a continuous flow fixed bed reactor under atmospheric pressure. Toluene (1000 ppm) in dry air was continuously supplied to the fixed bed reactor. The total flow rate and gas hourly space velocity were 200 mL $min^{-1}$ and 60,000 mL $g^{-1}$ $h^{-1}$ (0.2 g catalyst, 250–500 μm), respectively. Gas chromatography (GC-7890A, Agilent Technologies, Santa Clara, CA, USA) with a flame ionization detector and GS-GASPRO capillary column was used to identify the gases flowing in and out of the reactor. The measurements were performed at various temperatures ($T_{10}$, $T_{50}$, and $T_{90}$) corresponding to toluene conversions of 10, 50, and 90%, respectively, to evaluate the catalytic activity. The toluene conversion was calculated using Equation (1):

$$\text{Toluene conversion (\%)} = \frac{(C_{in} - C_{out})}{C_{in}} \times 100 \qquad (1)$$

where $C_{in}$ and $C_{out}$ are the toluene concentrations in the inlet and outlet gases, respectively. Each reaction rate was obtained from a toluene conversion below 20% to eliminate the

effects of heat and mass transfer [41]. The reaction rate ($r_{tol}$) and TOF for toluene oxidation were calculated using Equations (2) and (3):

$$r_{tol} \left( \text{mol s}^{-1} \text{g}_{\text{cat}}{}^{-1} \right) = \frac{X_{tol} F_{tol}}{m_{cat}} \tag{2}$$

$$\text{TOF} \left( \text{s}^{-1} \right) = \frac{r_{tol} M_{pt}}{X_{Pt} D_{Pt}} \tag{3}$$

where $X_{tol}$, $F_{tol}$, $m_{cat}$, $M_{Pt}$, $X_{Pt}$, and $D_{Pt}$ are the toluene conversion, molar flow rate of toluene, catalyst weight, atomic weight of Pt (195.084 g mol$^{-1}$), Pt content of the Pt/Y zeolites, and Pt dispersion measured by CO chemisorption, respectively.

The apparent activation energy was determined using the slope of the Arrhenius equation (Equation (4)):

$$\ln \text{k} = -\frac{E_a}{RT} + \ln A \tag{4}$$

where $E_a$, $T$, and $A$ are the apparent activation energy, reaction temperature, and pre-exponential factor, respectively.

## 4. Conclusions

Hierarchical Pt/Y zeolites with tunable mesopores were synthesized via the surfactant-templated crystal rearrangement method by varying the etching time. The hierarchical Pt/Y zeolites with abundant mesopores positively affected the Pt nanoparticle size, dispersion, Pt oxidation state, and surface acidity. Consequently, hierarchical Pt/Y zeolite catalysts exhibited higher activity than that observed using the non-hierarchical Pt/Y zeolite in the catalytic oxidation of toluene. Among the hierarchical Pt/Y zeolites, Pt/Y-6h exhibited the highest catalytic activity with a T$_{90}$ of 149 °C, reaction rate of 1.15 × 10$^{-7}$ mol g$_{\text{cat}}{}^{-1}$ s$^{-1}$, TOF of 1.20 × 10$^{-2}$ s$^{-1}$, and $E_a$ of 66.5 kJ mol$^{-1}$ at 60,000 mL g$^{-1}$ h$^{-1}$ with a toluene concentration of 1000 ppm. Based on a comparison of the reported Pt supported catalysts for toluene oxidation, as listed in Table 5, Pt/Y-6h is one of the highest-performing catalysts in this regard. This study therefore demonstrates the importance of introducing mesopores to improve the activity of Pt-supported catalysts. In addition, the methodology presented herein will pave the way for the fine-tuning of hierarchically porous materials with improved material characteristics and catalytic performances toward several different reactions.

**Table 5.** Performance comparison of the Pt-supported catalysts for toluene oxidation.

| Catalysts | Toluene Concentration (ppm) | GHSV [a] (mL g$^{-1}$ h$^{-1}$) | T$_{50}$ (°C) | T$_{90}$ (°C) | Pt Content (wt%) | Reference |
|---|---|---|---|---|---|---|
| Pt/Y-6h | 1000 | 60,000 | 141 | 149 | 0.41 | This study |
| Pt/Hβ | 1000 | 60,000 | - | 178 | 0.59 | [42] |
| Pt-G/@Zr | 1000 | 60,000 | 160 | 172 | 0.57 | [43] |
| Pt@PZN-2 | 1000 | 60,000 | 171 | 176 (T$_{98}$) | 0.50 | [44] |
| Ptnano/CoOx | 1000 | 30,000 | 165 | 177 | 1.00 | [45] |
| Pt–Co(OH)2–O | 1000 | 60,000 | 157 | 167 | 0.96 | [46] |

[a] Gas hourly space velocity.

**Author Contributions:** Conceptualization, M.-R.K. and S.K.; methodology, M.-R.K. and S.K.; validation, M.-R.K. and S.K.; investigation, M.-R.K. and S.K.; resources, M.-R.K. and S.K.; data curation, M.-R.K. and S.K.; writing—original draft preparation, M.-R.K. and S.K.; writing—review and editing, M.-R.K. and S.K.; visualization, M.-R.K. and S.K.; supervision, S.K.; project administration, S.K.; funding acquisition, S.K. All authors have read and agreed to the published version of the manuscript.

**Funding:** This study was conducted with the support of the Korea Institute of Industrial Technology as "Development of eco-friendly production system technology for total periodic resource cycle (KITECH EO-22-0007)".

**Data Availability Statement:** Not applicable.

**Conflicts of Interest:** The authors declare no conflict of interest.

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
