# Peer review of "Enhanced Catalytic Oxidation of Toluene over Hierarchical Pt/Y Zeolite"

_catalysts, doi:10.3390/catal12060622_

Round 1

Reviewer 1 Report

The manuscript (Enhanced catalytic oxidation of toluene over hierarchical Pt/Y 2 zeolite) reported an interesting mesoporous zeolite as advanced catalyst support for Pt catalyst, which demonstrated a superior catalytic activity for toluene oxidation. I think the results will be meaningful for the further development of supported catalyst for VOCs abatement. Minor revision is suggested.

Comment 1. Some peaks are not identified in the XRD pattern such as the peak at around 30o. Please make explanations.

Comment 2. The author used BJH method to evaluate the pore size distribution. However BJH is more suited for the analysis of mesopore structure, NL-DFT mode is suggested.

Comment 3. There is problem in Fig. 2b and Fig. 6a, please make modifications.

Comment 4. Please make more analysis for the significance of surface acidity for the improvement of the catalytic activity. Why the acidic sites help improve the toluene activation.

Comment 5. I noticed that there is no water involved in the reaction test, which is in contrast to the practical situation. Please make comments on the water resistance of the zeolite supported Pt catalyst.

Comment 6. The following manuscript is suggested for the citation. (Fuel, 2022, 214, 122774, doi:10.1016/j.fuel.2021.122774)

Author Response

Thank you for your kind comments. We corrected some broken fonts in figures and added explanation about acidity effect on catalytic activity. Also suggested paper was cited on our manuscript. In the revised manuscript, the red color is used for the changed parts.

Reviewer 2 Report

In general, the manuscript of Min Ryeong and Suhan Kim is well prepared and presented in terms of the methodology and results. The study covers a topic of relevance for the application of zeolites, not only for the oxidation of toluene, but also in catalysis in general. However, before recommending its publication in Catalysts, the authors must cover the points mentioned in the attached file. This work must be reconsidered after major revision.

Author Response

Thank you for your helpful comments. Based on your comments, we corrected broken words on figures and add more explanation about BET surface area change in the manuscript. We have added explanations about acidity change and a new table of acidity properties of hierarchical zeolites. Also catalytic performance comparison with previous studies was listed in table 5. In the revised manuscript, the red color is used for the changed parts.

Reviewer 3 Report

Kim et al. investigated the effect of etching time on the fabrication of mesopores into Pt/Y zeolite. The authors provided the influence of mesopores on Pt size, dispersion, available acidic site, and overall effect on Toluene oxidation. The manuscript's current condition would need revisions before it could be considered for publication. The authors' work can be considered for publication if they make the following changes.

  • Since the formation of mesopore is monitored, authors need to provide the small-angle XRD (0-5, 2theta) as well.
  • Line 86-88: Need to add zoom-in XRD spectra for the Pt peaks at 39.8 and 46.2 as inset. Also, mark the metal peaks (if possible) with asterisks to distinguish them in Fig 1.
  • Which model is utilized to study the Pore size distribution? BJH? Need to mention it in the title for Fig 2b.
  • Table 1: There is a slight drop in the surface area which could be a result of etching. This needs to be mentioned.
  • Table 1: How is the Vmeso rate calculated? Needs to be mentioned in the caption below.
  • Table 2, Pt particle size: How many particles were measured for this? what about the Pt particle size from CO chemisorption? are the sizes consistent obtained from both measurements?
  • NH3-TPD: The additional peak at higher temperature in the case of Pt/Y-6h indicates the presence of stronger acidic site types. However, the author needs to calculate the number of acidic sites (mmol/g) from calibration for each material.
  • Fig 6a: X-axis title needs to be corrected
  • For Arrhenius plots were kinetic reactions carried out to obtain these rates? if yes, please add the kinetic data points.
  • Based on the discussion and conclusion, the authors suggest that the mesopores help in confining the size of Pt particles? Need to discuss it and cite similar work (Jihong Yu Advanced Material 2018, Igor Slowing ChemCatChem 2021, Philippe Serp ChemCatChem2013).

Author Response

Thanks for your helpful comments. Upon your kind suggestion, we added Zoom-in-XRD spectra at Pt peak position at figure 1 inset. Explanation on surface area change was added in the manuscript. Also caption of table 1 is revised to describe how we calculated pores. Very helpfully Pt particle size can be confirmed by CO chemisorption data. And Pt particle size calculated by CO chemisorption data was added in Table 2. The number of acidic sites were calculated and added in table 3. For the Arrhenius plot, we did not carried out kinetic experiments, we assumed that the catalytic oxidation of toluene in air is a first-order reaction. Suggested paper was also cited on our manuscript. In the revised manuscript, the red color is used for the changed parts.

Reviewer 4 Report

The manuscript is devoted to the topical problem of the development of the platinum catalysts for the removal of volatile organic compounds.

The manuscript is well written. Results are interesting. At the same time there are some comments and questions:

1)   XPS is very important for the conclusions made in work: What software was used for data analysis? This is important for Pt state identification. How Pt4f + Al2p deconvolution was done? (Usually Al2p and Al2s regions for clean support are involved.) What line and at what position was used for internal calibration. This is important for Pt state identification. Generally accepted: Pt(met) BE 71.2 eV.

71.8 eV is the range of size effects of Pt (small particles and clusters). Are you sure that you do not have bimodal size distribution?

Just by eye: Pt4f7/2 peak area is less than Pt4f5/2. And the ratio looks conversely to the theory. Please, check and confirm your deconvolution before publication! See J.F. Moulder, W.F. Stickle, P.E. Sobol, K.D. Kenneth, Handbook of X-ray photoelectron spectroscopy, Ed.: Chastain J., King R.C.Jr., Physical Electronics, Inc., 1995.

Area for Pt4f7/2 : Pt4f5/2 = 8 : 6

What about FWHM for Al2p for different samples? Looks different…

2)   According to NH3 desorption and toluene oxidation reaction activity Pt-2h and Pt-4 h samples are rather similar. At the same time, the Pt-6h sample is differ. Could you explain such drastic alteration of sample behavior when the time is increased from 4 h to 6 h? Is 6 h the optimal time? Or 8 h sample could be better? Did you check?

3)   The activation energy for toluene oxidation reaction for different samples is noticeably different. What was the reason for the choice of the temperature for the plot for sample Pt-6h?  For the other samples plots are in the range of 130-150C, for the Pt-6h sample the range is 110-130C. Could you please show the data for all samples in the same temperature range? It looks like if you exclude the point with the lowest temperature activation energy for all samples will be noticeably closer.

4)   What products of toluene oxidation do you see? Only CO2? Or CO2, CO and other products of partial oxidation? Does the coke formed under used conditions? Do the catalysts stable during time?

5)   It is well known that Pt nanoparticles can sinter during reaction forming large particles. It would be interesting to see particle size distribution for Pt-6h sample after reaction.   

Since the manuscript is very short, there are a lot of place for catalysts stability and deactivation investigation as well as discussion of the origins of the shown effects.  

Manuscript cannot be published with present XPS spectra deconvolution. 

Author Response

Thanks for your helpful comments. Upon your comment, XPS data were rechecked and deconvolution was carried out again. There’s a little differences after re-deconvolution. Pt0 / (Pt0 + Pt2+) value in table 2 was corrected. A comment on CO2 selectivity was added on the manuscript shortly. In the revised manuscript, the red color is used for the changed parts. Replies to other comment are attached as a file.

Round 2

Reviewer 3 Report

I went through the author's response and the author did incorporate all the changes and has answered all the comments. I accept the paper for publication without any additional revision.

Author Response

Thanks for your acceptance of our paper.

Reviewer 4 Report

It would be nice to add the data about coke formation and catalyst deactivation.

About XPS: Looks much better. But, typically BE is higher for Pt2+. And pay attention, that by eye Pt2+/Al atomic ratio is close for all samples. So, it can be assumed that you have sintering of Pt0 (Pt4f line intensity drops), and Pt2+ is stable to sintering. And this can be conformation of the idea about Pt2+.

Manuscript can be published, but analysis of Pt2+/Al atomic ratios should be done and added.

Author Response

As you suggested, we added sentences about the coke formation. 
